# What fraction of cellular DNA turnover becomes cfDNA?

Ron Sender[1], Elad Noor[1], Ron Milo[1]*, Yuval Dor[2]*

[1]Weizmann Institute of Science, Rehovot, Israel; [2]Department of Developmental Biology and Cancer Research, Institute for Medical Research Israel Canada, the Hebrew University-Hadassah Medical School, Jerusalem, Israel

**Abstract** Cell-free DNA (cfDNA) tests use small amounts of DNA in the bloodstream as biomarkers. While it is thought that cfDNA is largely released by dying cells, the proportion of dying cells' DNA that reaches the bloodstream is unknown. Here, we integrate estimates of cellular turnover rates to calculate the expected amount of cfDNA. By comparing this to the actual amount of cell type-specific cfDNA, we estimate the proportion of DNA reaching plasma as cfDNA. We demonstrate that <10% of the DNA from dying cells is detectable in plasma, and the ratios of measured to expected cfDNA levels vary a thousand-fold among cell types, often reaching well below 0.1%. The analysis suggests that local clearance, presumably via phagocytosis, takes up most of the dying cells' DNA. Insights into the underlying mechanism may help to understand the physiological significance of cfDNA and improve the sensitivity of liquid biopsies.

## eLife assessment

This manuscript describes a model to estimate what fraction of DNA from specific human tissues becomes cell-free DNA in plasma. This **fundamental** study, supported by **convincing** evidence, will be of great interest to the community, as the amount of DNA from a certain tissue (for example, a tumor) that becomes available for detection in the blood has significant implications for disease detection.

*For correspondence:
ron.milo@weizmann.ac.il (RM);
yuvald@ekmd.huji.ac.il (YD)

Competing interest: The authors declare that no competing interests exist.

## Introduction

The human body is in a constant state of cellular turnover, with an estimated $0.3 \times 10^{12}$ cells being replaced daily, two-thirds of which are erythrocytes (*Sender and Milo, 2021*). Apoptosis is the primary mechanism of cell death in tissues with rapid turnover, in which cells are disposed of orderly and engulfed by phagocytic cells to recycle their resources.

Apoptosis involves the fragmentation of DNA through two distinct mechanisms. The first mechanism occurs within the apoptotic cell, where endonucleases break down chromatin into nucleosomal units. The second mechanism is carried out by phagocytic cells, which engulf and degrade the DNA of apoptotic cells in order to prevent the release of potentially immunogenic intracellular materials (*Nagata, 2005*).

It has been known since the late 1940s that cell-free DNA (cfDNA) fragments can be found in the circulation of healthy and diseased individuals (*Mandel and Metais, 1948*; *Bendich et al., 1965*). The fragments are typically nucleosome-size (165 base pairs), likely representing molecules that were not completely degraded during the process of cell death. With the advent of next-generation sequencing (NGS), cfDNA has become a clinically useful biomarker for various applications. These include non-invasive prenatal testing (via detection of fetal chromosomal abnormalities via sampling of maternal cfDNA), cancer monitoring (via detection of oncogenic mutations in plasma, termed circulating tumor

DNA [ctDNA]), and monitoring of allogeneic organ transplants (via detection of donor-derived SNPs in cfDNA)(*Heitzer et al., 2019*). Perhaps, the greatest promise of cfDNA is in cancer diagnostics – a blood test that can allow early detection at an actionable stage, real-time assessment of treatment response, detection of recurrence, and identification of specific genetic mutations to inform treatment decisions (*Jamshidi et al., 2022*; *Wan et al., 2017*).

Regardless of somatic mutations, the mere presence of cfDNA from a given tissue is of great value as it often correlates with tissue-specific injury (*Lehmann-Werman et al., 2016*; *Gala-Lopez et al., 2018*; *Lehmann-Werman et al., 2018*; *Zemmour et al., 2018*; *Heitzer et al., 2020*). Multiple layers of epigenetic information allow the inference of the tissue origins of cfDNA. For example, the size, fragmentation patterns, and exact end position of cfDNA fragments, nucleosome positions reflected in the relative abundance of promoter sequences, and histone modification patterns all allow tracing cfDNA molecules to their tissue origin (*Moss et al., 2018*; *Oberhofer et al., 2022*; *Zhou et al., 2022*; *Lo et al., 2021*). One particularly sensitive approach is using DNA methylation patterns, a stable determinant of cell identity preserved on cfDNA. Deconvolution of cfDNA methylomes using a reference atlas of human cell-type-specific methylomes has revealed various tissues' relative and absolute contribution to cfDNA in health and disease. Under baseline conditions, over 90% of cfDNA originates in blood cells (neutrophils, megakaryocytes, monocytes, lymphocytes, and erythroblasts), with vascular endothelial cells and hepatocytes being the only solid tissue source (*Loyfer et al., 2023*). In both homeostatic and pathologic conditions, the exact mechanism by which cfDNA is released is not fully understood but is thought to involve cell death. Whether cfDNA can be released from cells that remain alive after the event is controversial (*Stroun et al., 2001*). Two striking examples of such a scenario are megakaryocytes and erythroblasts, whose physiological function is to release anuclear cells, namely platelets and erythrocytes (*Moss et al., 2022*).

From a practical perspective, the amount of cfDNA (typically ~1000 genome equivalents per ml of plasma) is a major barrier to a sensitive diagnosis of diseases – particularly cancer – at an early stage. Beyond maximization of the volume of blood drawn and the number of markers tested in parallel, understanding and eventually manipulating the local release and systemic clearance of cfDNA hold great potential for improving the sensitivity of tests. For example, recent studies have suggested pharmacologic approaches for blocking the removal of cfDNA from the systemic circulation, leading to a transient elevation in cfDNA concentration (*Tabrizi et al., 2023*). The efficiency and determinants of local cfDNA release to circulation have not been examined.

In this study, we use recent estimates of cellular turnover rates (*Sender and Milo, 2021*) to calculate the expected amount of DNA resulting from cell death from each cell type at a given time. By comparing this to the amount of cell type-specific cfDNA present in the plasma, taking into account estimates of systemic cfDNA clearance rate, we estimate the fraction of DNA that reaches the plasma as cfDNA.

## Results

Based on atlases of human cell-type-specific methylation signatures, Moss et al. and Loyfer et al. analyzed the main cell types contributing to plasma cfDNA. They found the primary sources of plasma cfDNA to be blood cells: granulocytes, megakaryocytes, macrophages, and/or monocytes (the signature could not differentiate between the last two), lymphocytes, and erythrocyte progenitors. Other cells that had detectable contributions are endothelial cells and hepatocytes. Qualitatively, these cells represent most of the leading cell types in cellular turnover, as shown in *Sender and Milo, 2021*. Epithelial cells of the gastrointestinal tract, lung, kidney, bladder, and skin are other cell types that significantly contribute to cellular turnover. Dying cells in these tissues are shed into the gut lumen, the air spaces, the urine, or out of the skin (note that while DNA from gut, lung, and kidney epithelial cells can be found in stool, bronchoalveolar lavage, and urine, the fate of DNA from skin cells is not known). This arrangement may explain why DNA from these cell types is not represented in plasma cfDNA in healthy conditions. Therefore, it appears that cells with high cfDNA plasma levels are those with relatively high turnover that are not being shed out of the body.

We used the cellular turnover estimates of these cell types to calculate the potential amount of DNA discarded from each cell type. We derived the potential levels of cfDNA in the plasma (see Materials and methods). By comparing this data to the measured levels of cfDNA in plasma (*Moss et al., 2018*; *Loyfer et al., 2023*), we could calculate the fraction of potential DNA presented as cfDNA in

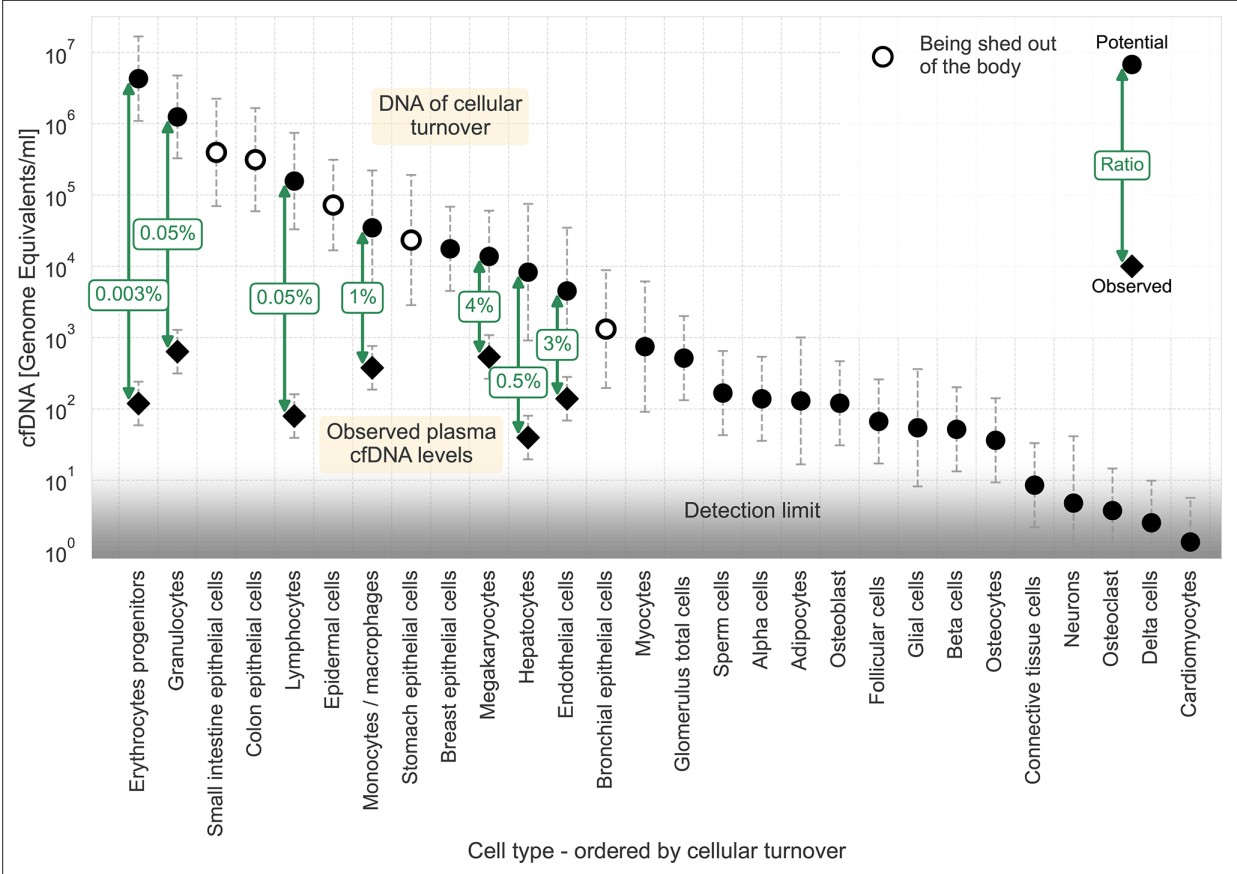

**Figure 1.** cfDNA as a fraction of homeostatic cell turnover. Estimates for the DNA flux from homoeostatic cell turnover were made based on *Sender and Milo, 2021* and converted to units of potential cfDNA plasma concentration (see Methods). Cell types are ordered by their estimated cellular turnover. Empty markers represent cell types being shed out of the body upon turnover. Observed levels of cfDNA (*Moss et al., 2018*; *Loyfer et al., 2023*) are shown for cell types found in the plasma, with labels depicting the ratio between potential and measured cfDNA concentrations. Circles, potential representation in cfDNA; diamonds, observed concentration of cfDNA. The assay's detection limit is presented by a gradient (from around ten genome equivalents for deconvolution assay to around one genome equivalent for targeted assays). Error bars represent the 95% confidence interval of the uncertainty, approximated by a lognormal distribution (see Materials and methods).

the plasma, as illustrated in *Figure 1*. The results indicate that less than 4% of the DNA of dying cells reaches the plasma. The ratios of measured to expected cfDNA levels vary a thousand-fold, ranging from 1:30 (megakaryocytes and endothelial cells) to $1:3\times10^4$ (erythrocyte progenitors).

In general, around 1000 genome equivalents of cfDNA are found in the plasma in healthy individuals. The limit for detection of a cell type of specific origin depends on the assay in use. General essays using deconvolution have a sensitivity of around 1%, for example, ~10 genome equivalents (*Loyfer et al., 2023*). Targeted essays using markers for a specific cell type at a deep coverage can improve the sensitivity to around 0.1%, for example ~1, genome equivalent. The gradient in *Figure 1* depicts this range of sensitivities. The low ratios of measured to potential cfDNA described for the mentioned cell types indicate that cells with lower cellular turnover, such as skeletal myocytes, adipocytes, and pancreatic beta cells, are not being detected in the plasma of healthy individuals because their plasma levels are lower than the sensitivities of existing essays. Notably, a comparison of potential cfDNA plasma levels of breast epithelial cells in healthy women to the limit of detection reveals the breast as an outlier. This might suggest that dying breast epithelial cells' local DNA utilization mechanism is extremely efficient.

## Discussion

In this study, we report a surprising, dramatic discrepancy between the measured levels of cfDNA in the plasma and the potential DNA flux from dying cells. One hypothetical explanation for that discrepancy is the limited sensitivity of typical cfDNA assays to short DNA fragments, which may contribute a significant fraction of the overall cfDNA mass. Regular cfDNA analysis shows a size distribution concentrated around a length of 165 base pairs (bp). The sizes in ctDNA vary more, but most are longer than 100 bp (*Alcaide et al., 2020*; *Udomruk et al., 2021*). Recent studies suggested a significant fraction of single-strand ultrashort fragments (length of 25–60 bp; *Cheng et al., 2022*; *Hisano et al., 2021*). However, the total amount of DNA contained in these fragments is less than or comparable to that of the longer 'regular' nucleosome-protected cfDNA fragments (*Cheng et al., 2022*; *Hisano et al., 2021*), arguing against ultrashort fragments as a dominant explanation for the 'missing' cfDNA material. We integrated the estimate provided by Hisano et al. into our analysis as a modifying factor for both the total concentration and uncertainty of plasma cfDNA. Importantly, this incorporation did not alter the overall conclusions, as the discrepancy between the cfDNA plasma concentration and potential DNA flux remains on the same order of magnitude. We note that we cannot exclude the presence of abundant DNA fragments that are even shorter (e.g. 10 bp long) and are not measurable in cfDNA analysis. Thus, our formal conclusion is that only a small fraction of the DNA of dying cells appears as measurable cfDNA.

An alternative hypothetical explanation is that most DNA of dying enters the bloodstream but is rapidly degraded or taken up. Accounting for the tissue-specific DNA concentration found in the blood, we can estimate the half-life of cfDNA in the bloodstream in that case, based on the cellular turnover rate. This calculation suggests that the half-life of cfDNA in the bloodstream should be only a few seconds to a few minutes. However, previous research using various methods (mostly the decay of fetal cfDNA in maternal plasma after birth) has shown that the half-life of DNA in the bloodstream ranges from 15 to 120 min, orders of magnitude higher than this estimate suggest (*Lo et al., 1999*; *Yao et al., 2016*; *Diehl et al., 2008*; *To et al., 2003*). In addition, a systemic clearance mechanism cannot explain the differential representation of cfDNA from different cell types relative to their turnover rate.

Therefore, the low fraction of DNA measured as cfDNA suggests that less than a few percent of the DNA from dying cells reaches the bloodstream. An unknown mechanism utilizes the rest with a tissue-specific efficiency. A potential explanation is that tissue-resident phagocytes are degrading the DNA after apoptosis, similar to what has been shown for extruded erythroblast nuclei (*Yoshida et al., 2005*). Studies in mice have revealed that lysosomal DNases in macrophages play a role in cleaving chromosomal DNA during apoptosis (*McIlroy et al., 2000*), as well as the degradation of erythroid nuclei in erythroblastic islands (*Yoshida et al., 2005*). The first DNA fragmentation occurs within the apoptotic cells, resulting in nucleosomal unit multiples of about 180 bp (*Matassov et al., 2004*). Therefore, up to 200 bp cfDNA fragments in plasma may indicate that phagocytes have not further degraded these fragments.

Our current analysis focused on estimating plasma cfDNA concentration and cellular turnover in a cohort of healthy, relatively young individuals. The total plasma cfDNA concentrations were sourced from healthy individuals below 47 years, as reported by *Meddeb et al., 2019*. We use data analyzed based on plasma samples from healthy individuals to estimate the proportion of cfDNA originating from specific cell types (*Loyfer et al., 2023*). These values were then compared to the potential DNA flux resulting from homeostatic cellular turnover, estimated for reference healthy males aged between 20 and 30 (*Sender and Milo, 2021*). In our analysis, we considered various sources of uncertainty, including inter-individual variation, variability in the timing of sample collection, and analytical precision (*Madsen et al., 2019*; *Meddeb et al., 2019*). These factors collectively contributed to an uncertainty factor of less than 3. Importantly, this level of uncertainty does not alter our conclusion regarding the relatively small fraction of DNA present in plasma as cfDNA. Furthermore, we acknowledge that age and sex can impact total cfDNA concentration, as demonstrated by *Meddeb et al., 2019*, with potential variations of up to 30%. However, as the results of our analysis present a much larger difference, these effects do not change the conclusions drawn from our analysis. Nevertheless, age and health status may influence the proportion of cfDNA originating from specific cell types and their corresponding cellular turnover rates. Consequently, the ratios themselves may vary in the elderly population or individuals with underlying health conditions.

A comparison between the different types of cells shows a trend in which less DNA flux from cells with higher turnover gets to the bloodstream. In particular, a tiny fraction (1 in $3 \times 10^4$) of DNA from erythroid progenitors arrives at the plasma, indicating an extreme efficiency of the DNA recovery mechanism. Erythroid progenitors are arranged in erythroblastic islands. Up to a few tens of erythroid progenitors surround a single macrophage that collects the nuclei extruded during the erythrocyte maturation process (pyrenocytes; *Chasis and Mohandas, 2008*). The amount of DNA discarded through the maturation of over 200 billion erythrocytes per day (*Sender and Milo, 2021*) exceeds all other sources of homeostatic discarded DNA. Our findings indicate that the organization of dedicated erythroblastic islands functions highly efficiently regarding DNA utilization. Neutrophils are another high-turnover cell type with a low level of cfDNA. When contemplating the process of NETosis (*Vorobjeva and Chernyak, 2020*), the existence of cfDNA originating from live neutrophils would potentially diminish the concentration of cfDNA released by dying neutrophils, thereby amplifying the observed ratio for this particular cell type. The overall trend of higher turnover resulting in a lower cfDNA to DNA flux ratio may indicate similar design principles, in which the utilization of DNA is better in tissues with higher turnover. However, our analysis is limited to only several cell types (due to cfDNA test and deconvolution sensitivities), and extrapolation to cells with lower cell turnover is problematic.

A thorough explanation for the gap between the estimated DNA flux from dying cells and the measured cfDNA data requires more research. Since macrophages play a prominent role in the phagocytosis of dying cells, we hypothesize that the local uptake of cfDNA by activated macrophages is responsible for the uptake of most DNA from dying cells. An interesting implication of this possibility is that cfDNA levels are expected to be highly sensitive to perturbations in the local clearance mechanism. In other words, elevated levels of cfDNA from a given cell type may represent a disruption of local macrophages rather than an actual increase in the rate of cell death.

Comparing the DNA flux involved in the homeostatic cellular turnover of specific cell types to the sensitivity of cfDNA essays reveals some current limitations in the field. Cell types such as adipocytes, cardiomyocytes, and pancreatic beta cells are not represented in the cfDNA of healthy individuals. Our analysis suggests that current essays need to be more sensitive to identify the minute amount of those. Moreover, the quantitative analysis can predict potential cell types with a non-neglectable contribution to the plasma cfDNA and allow their focused study. The current analysis calls for a focus on breast epithelial cells and myocytes, as their potential cfDNA levels are relatively higher than the detection limit. Previous research regarding their contribution to cfDNA has used highly sensitive essays but found no contribution in healthy individuals (*Moss et al., 2018*; *Moss et al., 2020*; *Loyfer et al., 2023*). This might indicate a highly effective mechanism for the utilization of DNA from dying cells, for example, local degradation of myonuclei within the syncytium of skeletal muscle fibers.

Quantitative characterization of the abundance of macrophages concerning the cellular death rate in different tissues could improve our understanding of the DNA clearance mechanism and the role of phagocytes.

Illuminating the discrepancy between the dying cells' DNA flux and the measured cfDNA levels may open the door for research with clinical potential. The sensitivity of the assays and the amount of available DNA limit the utility of liquid biopsy, particularly in early disease detection. Better characterization of the mechanism that limits available plasma cfDNA could lead to potential interventions that increase the fraction of DNA flux arriving at the plasma and thus improve the sensitivity of liquid biopsies based on cfDNA.

## Materials and methods
### Cellular turnover

Normal cellular turnover data for all cell types except hepatocytes and megakaryocytes were obtained from estimates provided by *Sender and Milo, 2021*. We estimated the cellular turnover of megakaryocytes in two ways. First, based on the number of megakaryocytes in the bone marrow (*Harrison, 1962*; *Noetzli et al., 2019*) and a maturation time of around five days (*Machlus and Italiano, 2013*). Second, using the production of platelets (*Harker and Finch, 1969*) and an average number of platelets produced per megakaryocyte (*Trowbridge et al., 1984*; *Kaufman et al., 1965*; *Harker and Finch, 1969*). Cellular turnover of hepatocytes was calculated based on *Heinke et al., 2022* by combining estimates for the number of cells and the death rate for the different ploidy groups (*Source data 1*).

## Tissue-specific cfDNA concentration

Our estimates for total plasma cfDNA concentration were derived from the median concentration observed in individuals below 47 years of age (n=52), as reported by *Meddeb et al., 2019*. To complement this, we integrated our total concentration estimates with data on the proportion of cfDNA originating from specific cell types, leveraging a plasma methylome deconvolution method described by *Loyfer et al., 2023*, which did not provide absolute quantities of cfDNA.

The overall plasma cfDNA concentration was multiplied by a factor of 1.5 to accommodate for the presence of small fragments of approximately 50 base pairs of cfDNA in the plasma. These fragments are suggested to contribute comparable molar concentrations (*Hisano et al., 2021*). Despite having approximately one-third of the mass, it is reasonable to presume that these fragments represent a similar number of genomes. This assumption is based on the idea that their source is a broken nucleosome unit, and the fragments represent the portion that was not degraded. Given the restricted data and its interpretation, we consider factors spanning the range of 1 (negligible effect) and 2 (doubling of the amount). The chosen factor, 1.5, is selected as the midpoint within this range of uncertainty.

To quantify the uncertainty associated with our cfDNA concentration estimates, we employed a methodology that considered several sources of variation. First, we incorporated the confidence interval of the median concentration reported by Meddeb et al. as a measure of uncertainty. Additionally, we accounted for individual-specific and analytic variations based on the study by *Madsen et al., 2019*, encompassing factors such as the precise timing of measurements and assay precision. These sources of uncertainty were combined using the approach outlined below.

## Estimation of the potential DNA flux

We estimated the potential cfDNA plasma levels if all the DNA from the dying cells had reached the bloodstream. Our estimate utilized the calculated cellular turnover rate and data regarding the ploidy of the cells, the volume of blood plasma, and the half-life of cfDNA molecules in the blood. For each cell type, we defined the cellular turnover in units of cells per day $d_c$ and the ploidy (average number of sets of chromosomes) $p_c$ . We used blood plasma volume $V_{plasma} = 3L$ (*Valentin, 2002*), the mean lifespan of cfDNA molecules in the blood $\tau = \frac{half-life}{ln(2)} = 0.7h$ (*Lo et al., 1999*; *Yao et al., 2016*; *Diehl et al., 2008*; *To et al., 2003*) and haploid genome mass $m_h = 3.2 \cdot 10^{-12} g$ (*Piovesan et al., 2019*).

The expected level of cfDNA levels was calculated according to the formula:

$$X_c = \frac{d_c \cdot p_c \cdot \cdot \tau}{V_{plasma}} \tag{1}$$

where $X_c$ is given in units of Genome equivalents/ml (Units of g/ml could be obtained by multiplication by $m_h$).

Intuitively, the potential cfDNA level is obtained by calculating the amount of DNA in dying cells at a given moment (defined by the mean lifespan of cfDNA in plasma) when considering the total volume of the plasma.

Ultimately, we compared the measured cfDNA levels to the potential DNA flux estimates to determine the DNA fraction reaching the blood.

## Uncertainties estimate

Standard error was collected or calculated for each value used. In several cases, such as the half-life of plasma cfDNA, the value's uncertainty was big and best described as a multiplication factor of error (i.e. the uncertainty of a variable with lognormal distribution). To facilitate error propagation, we transformed all values and corresponding errors to be expressed in terms of multiplication error by fitting a lognormal distribution. Thus, we modeled the uncertainty around the median value as a lognormally distributed random variable $x$ with a with a shape parameter $s = ln(f_x,)$, where $f_x$ is the multiplication error factor. For example, an $f_x = 2$ means that there is a 68% probability (one sigma) that the actual value of $x$ is between half and double the given value. The shape parameter describes the standard error of the log-transformed random variable, defined as $ln(x)$, and is distributed normally by definition.

The multiplication of two lognormal variables $x$ and $y$ also follows a log-normal distribution, with a shape parameter equal to the square root of the sum of the squares of the original shape factors.

Therefore, to propagate the error of the product $(x \cdot y)$, we used the formula: $f_{x \cdot y} = e^{\sqrt{ln(f_x)^2 + ln(f_y)^2}}$.

We used bootstrapping to calculate error propagation for the summation of variables with log-normal uncertainty (lacking an analytical formula for error propagation). Specifically, we drew 1000 samples from the distribution that described the uncertainties of the values.

The overall concentration of cfDNA in plasma was adjusted to account for the ubiquity of small cfDNA fragments. Based on Hisano et al. results, we assumed that the uncertainty associated with this correction could be represented by a uniform distribution of the adjustment factor within the range of 1–2 (see Tissue-specific cfDNA concentration). To incorporate this uncertainty into the total plasma cfDNA concentration estimate, we employed bootstrapping with 10,000 iterations. Subsequently, a lognormal distribution was fitted to the results to facilitate further error propagation.

## Acknowledgements

We thank Yinon Bar-On, Lior Greenspon, and Yuval Rosenberg for valuable feedback on this manuscript. Funding: This research was generously supported by the Mary and Tom Beck Canadian Center for Alternative Energy Research, the Schwartz-Reisman Collaborative Science Program, the Ullmann Family Foundation, and the Yotam Project (RM). This research was supported by grants from the Helmsley Charitable Trust, JDRF, NIDDK, and Grail (YD). Prof. Yuval Dor has filed patents on cfDNA analysis. Prof. Ron Milo is the Head of the Mary and Tom Beck Canadian Center for Alternative Energy Research and the Charles and Louise Gartner Professorial Chair incumbent.

## Additional information

### Funding

| Funder | Grant reference number | Author |
|---|---|---|
| Schwartz-Reisman collaborative science program | | Ron Milo |
| Yotam project | | Ron Milo |
| Mary and Tom Beck Canadian Center for Alternative Energy Research | | Ron Milo |
| Ulman Foundation | | Ron Milo |
| Leona M. and Harry B. Helmsley Charitable Trust | | Yuval Dor |
| JDRF | | Yuval Dor |
| National Institute of Diabetes and Digestive and Kidney Diseases | | Yuval Dor |
| Grail | | Yuval Dor |

The funders had no role in study design, data collection and interpretation, or the decision to submit the work for publication.

### Author contributions

Ron Sender, Conceptualization, Data curation, Software, Formal analysis, Validation, Investigation, Visualization, Methodology, Writing – original draft, Writing – review and editing; Elad Noor, Conceptualization, Software, Formal analysis, Supervision, Validation, Investigation, Visualization, Methodology, Writing – original draft, Writing – review and editing; Ron Milo, Conceptualization, Resources, Formal analysis, Supervision, Funding acquisition, Validation, Investigation, Visualization,

Methodology, Writing – original draft, Project administration, Writing – review and editing; Yuval Dor, Conceptualization, Resources, Data curation, Formal analysis, Supervision, Funding acquisition, Validation, Investigation, Visualization, Methodology, Writing – original draft, Project administration, Writing – review and editing

## Author ORCIDs
Ron Sender ⬦ https://orcid.org/0000-0002-1165-9818
Elad Noor ⬦ http://orcid.org/0000-0001-8776-4799
Ron Milo ⬦ http://orcid.org/0000-0003-1641-2299
Yuval Dor ⬦ https://orcid.org/0000-0003-2456-2289

Joint Public Review: https://doi.org/10.7554/eLife.89321.3.sa1
Author Response https://doi.org/10.7554/eLife.89321.3.sa2

---

# Additional files

### Supplementary files
• Source data 1. Summary of the data used for the comparison: cfDNA properties and cell-type specific cellular turnover data.

• MDAR checklist

### Data availability
All study data are included in the article and *Source data 1*. All code is available in Jupyter notebooks at https://gitlab.com/milo-lab-public/cfdna-and-cellular-turnover, copy archived at *Milo Lab, 2024*.

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
