## [Editor Report · eLife assessment]

This manuscript describes a model to estimate what fraction of DNA from specific human tissues becomes cell-free DNA in plasma. This **fundamental** study, supported by **convincing** evidence, will be of great interest to the community, as the amount of DNA from a certain tissue (for example, a tumor) that becomes available for detection in the blood has significant implications for disease detection.

---

## [Referee Report · Joint Public Review]

Summary

Sender et al describe a model to estimate what fraction of DNA becomes cell-free DNA in plasma. This is of great interest to the community, as the amount of DNA from a certain tissue (for example, a tumor) that becomes available for detection in the blood has important implications for disease detection.

Strengths

The question asked by the authors has potentially important implications for disease diagnosis. Understanding how genomic DNA degrades in the human circulation can guide towards ways to enrich for DNA of interest or may lead to unexpected methods of conserving cell-free DNA. Thus, the question "how much genomic DNA becomes cfDNA" is of great interest to the scientific and medical community. I believe this manuscript has the potential to be a widely used resource. As more data is collected on cell-free DNA yields and cellular turnover in the body, this work will only increase in importance.

Appraisal

At this stage of the manuscript (second submission), I think the authors provide important evidence and analysis that aim to answer their research question. Previous concerns about methodology have been addressed.

Impact

This manuscript will be highly impactful on the community. The field of liquid biopsies (non-invasive diagnostics) has the potential to revolutionize the medical field (and has already in certain areas, such as prenatal diagnostics). Yet, there is a lack of basic science questions in the field. This manuscript is an important step forward in asking more "basic science" questions that seek to answer a fundamental biological question.

---

## [Author Response]

The following is the authors’ response to the original reviews.

**Public Reviews:**

**Reviewer #1 (Public Review):**
Summary:Sender et al describe a model to estimate what fraction of DNA becomes cell-free DNA in plasma. This is of great interest to the community, as the amount of DNA from a certain tissue (for example, a tumor) that becomes available for detection in the blood has important implications for disease detection.However, the authors' methods do not consider important variables related to cell-free DNA shedding and storage, and their results may thus be inaccurate. At this stage of the paper, the methods section lacks important detail. Thus, it is difficult to fully assess the manuscript and its results.Strengths:The question asked by the authors has potentially important implications for disease diagnosis. Understanding how genomic DNA degrades in the human circulation can guide towards ways to enrich for DNA of interest or may lead to unexpected methods of conserving cell-free DNA. Thus, the question "how much genomic DNA becomes cfDNA" is of great interest to the scientific and medical community. Once the weaknesses of the manuscript are addressed, I believe this manuscript has the potential to be a widely used resource.Weaknesses:There are two major weaknesses in how the analysis is presented. First, the methods lack detail. Second, the analysis does not consider key variables in their model.Issues pertaining to the methods section.The current manuscript builds a flux model, mostly taking values and results from three previous studies:1. The amount of cellular turnover by cell type, taken from Sender & Milo, 20211. The fractions of various tissues that contribute DNA to the plasma, taken from Moss et al, 2018 and Loyfer et al, 2023My expertise lies in cell-free DNA, and so I will limit my comments to the manuscripts in (2).Paper by Loyfer et al (additional context):Loyfer et al is a recent landmark paper that presents a computational method for deconvoluting tissues of origin based on methylation profiles of flow-sorted cell types. Thus, the manuscript provides a well-curated methylation dataset of sorted cell-types. The majority of this manuscript describes the methylation patterns and features of the reference methylomes (bulk, sorted cell types), with a smaller portion devoted to cell-free DNA tissue of origin deconvolution.I believe the data the authors are retrieving from the Loyfer study are from the 23 healthy plasma cfDNA methylomes analyzed in the study, and not the re-analysis of the 52 COVID-19 samples from Cheng et al (MED 2021).Paper by Moss et al (additional context):Moss et al is another landmark paper that predates the Loyfer et al manuscript. The technology used in this study (methylation arrays) is outdated but is an incredible resource for the community. This paper evaluates cfDNA tissues of origin in health and different disease scenarios. Again, I assume the current manuscript only pulled data from healthy patients, although I cannot be sure as it is not described in the methods section.This manuscript:The current manuscript takes (I think) the total cfDNA concentration from males and females from the Moss et al manuscript (pooled cfDNA; 2 young male groups, 2 old male groups, 2 young female groups, 2 old female groups, Supplementary Dataset; "total_cfDNA_conc" tab). I believe this is the data used as total cfDNA concentration. It would be beneficial for all readers if the authors clarified this point.The tissues of origin, in the supplemental dataset ("fraction" tab), presents the data from 8 cell types (erythrocytes, monocytes/macrophages, megakaryocytes, granulocytes, hepatocytes, endothelial cells, lymphocytes, other). The fractions in the spreadsheet do not match the Loyfer or Moss manuscripts for healthy individuals. Thus, I do not know what values the supplementary dataset represents. I also don't know what the deconvolution values are used for the flux model.The integration of these two methods lack detail. Are the authors here using yields (ie, cfDNA concentrations) from Moss et al, and tissue fractions from Loyfer et al? If so, why? There are more samples in the Loyfer manuscript, so why are the samples from Moss et al. being used? The authors are also selectively ignoring cell-types that are present in healthy individuals (Neurons from Moss et al, 2018). Why?Appraisal:At this stage of the manuscript, I think additional evidence and analysis is required to confirm the results in the manuscript.Impact:Once the authors present additional analysis to substantiate their results, this manuscript will be highly impactful on the community. The field of liquid biopsies (non-invasive diagnostics) has the potential to revolutionize the medical field (and has already in certain areas, such as prenatal diagnostics). Yet, there is a lack of basic science questions in the field. This manuscript is an important step forward in asking more "basic science" questions that seek to answer a fundamental biological question.

We thank the reviewer for the valuable comments on our analysis. In response to the feedback, we have updated the analysis to address all critical points as described below and revised the text to enhance the clarity of our methodology. One notable improvement to our analysis involved ensuring better alignment between the cohort data for cfDNA plasma concentration and cell turnover estimates. To achieve this, we utilized the total plasma concentration of cfDNA from a study conducted by Meddeb et al. 2019, taking into account the influence of age and sex on these concentrations and specifically focusing on a cohort of relatively young and healthy individuals. Additionally, we considered expected variations related to sex, age, and other pertinent factors, as outlined in the studies by Meddeb et al. 2019 and Madsen et al. 2019.

In addition, we have addressed concerns regarding the technical aspects of cfDNA analysis, providing detailed explanations of their limited impact on our analysis and the resulting conclusions.

**Reviewer #2 (Public Review):**
Summary:Cell-free DNA (cfDNA) are short DNA fragments released into the circulation when cells die. Plasma cfDNA level is thought to reflect the degree of cell-death or tissue injury. Indeed, plasma cfDNA is a reliable diagnostic biomarker for multiple diseases, providing insights into disease severity and outcomes. In this manuscript, Dr. Sender and colleagues address a fundamental question: What fraction of DNA released from cell death is detectable as plasma cfDNA? The authors use public data to estimate the amount of DNA produced from dying cells. They also utilize public data to estimate plasma cfDNA levels. Their calculations showed that <10% of DNA released is detectable as plasma cfDNA, the fraction of detectable cfDNA varying by tissue sources. The study demonstrates new and fundamental principles that could improve disease diagnosis and treatment via cfDNA.Strengths:1. The experimental approach is resource-mindful taking advantage of publicly available data to estimate the fraction of detectable cfDNA in physiological states. The authors did not assess if the fraction of detectable cfDNA changes in disease conditions. Nonetheless, their pioneering study lays the foundation and provides the methods needed for a similar assessment in disease states.1. The findings of this study potentially explain discrepancies in measured versus expected tissue-specific cfDNA from some tissues. For example, the gastrointestinal tract is subject to high cell turnover and release of DNA. Yet, only a small fraction of that DNA ends up in plasma as gastrointestinal cfDNA.1. The study proposes potential mechanisms that could account for the low fraction of detectable cfDNA in plasma relative to DNA released. This includes intracellular or tissue machinery that could "chew up" DNA released from dying cells, allowing only a small fraction to escape into plasma as cfDNA. Could this explain why the gastrointestinal track with an elaborate phagosome machinery contributes a small fraction of plasma cfDNA? Given the role of cfDNA as damage-associated molecular pattern in some diseases, targeting such a machinery may provide novel therapeutic opportunities.Weaknesses:In vitro and in vivo studies are needed to validate these findings and define tissue machinery that contribute to cfDNA production. The validation studies should address the following limitations of the study design: -1. Align the cohorts to estimate DNA production and plasma cfDNA levels. Cellular turnover rate and plasma cfDNA levels vary with age, sex, circadian clock, and other factors (Madsen AT et al, EBioMedicine, 2019). This study estimated DNA production using data abstracted from a homogenous group of healthy control males (Sender & Milo, Nat Med 2021). On the other hand, plasma cfDNA levels were obtained from datasets of more diverse cohort of healthy males and females with a wide range of ages (Loyfer et al. Nature, 2023 and Moss et al., Nat Commun, 2018).1. "cfDNA fragments are not created equal". Recent studies demonstrate that cfDNA composition vary with disease state. For example, cfDNA GC content, fraction of short fragments, and composition of some genomic elements increase in heart transplant rejection compared to no-rejection state (Agbor-Enoh, Circulation, 2021). The genomic location and disease state may therefore be important factors to consider in these analyses.1. Alternative sources of DNA production should be considered. Aside from cell death, DNA can be released from cells via active secretion. This and other additional sources of DNA should be considered in future studies. The distinct characteristics of mitochondrial DNA to genomic DNA should also be considered.

We appreciate the reviewer's comments on our analysis. In response to the feedback, we have updated to address key points and revised the text accordingly.

1. We have incorporated several enhancements to improve the coherence of our analysis. In our revised examination, we drew upon the total plasma concentration of cfDNA, as documented in a study conducted by (Meddeb et al. 2019), while considering the influence of age and sex on these concentrations. To ensure the cohort's alignment, we focus on relatively young and healthy individuals, specifically those below the age of 47. This approach allowed for a more meaningful comparison with the estimated DNA flux from a reference male human aged between 20 and 30 years.

There was no specific estimate for a cohort of young males in both Meddeb et al. and Loyfer et al.; however, we factored in the expected variations stemming from sex, age, and other relevant factors, as elucidated in literature (Meddeb et al. 2019; Madsen et al. 2019). Thus, we demonstrate that sex and age have a small effect on the cfDNA concentrations and thus are unlikely to alter our conclusions substantially when considering a healthy population.We summarize the changes in the first paragraph, replacing the “Tissue-specific cfDNA concentration” subsection of the method, and the fourth paragraph added to the discussion.

1. In this study, we addressed the total amount of cfDNA in healthy individuals without regard to GC content, representation of different genomic regions, or fragment length, as the goal was to understand if cell death rates are fully accounted for by cfDNA concentration. We agree that it will be interesting to study the relative representation of the genome in cfDNA and the processes that determine cfDNA concentration in pathologies beyond the rate of cell death. These topics for future research fall beyond this study's scope.

2. We know only a few specific cases whereby DNA is released from cells that are not dying. These include the release of DNA from erythroblasts and megakaryocytes to generate anucleated erythrocytes and platelets (Moss et al. 2022, cited in our paper) and the release of NETs from neutrophils.

The presence of cfDNA fragments originating from megakaryocytes and erythroblasts indicates the elimination of megakaryocytes and erythroblasts and the birth of erythrocytes and platelets. However, the considerations in the rest of the paper still apply: the concentration of cfDNA from these sources is far lower than expected from the cell turnover rate.

Concerning NETosis: the presence of cfDNA originating in neutrophils that have not died would reduce the concentration of cfDNA from dying neutrophils and thus further increase the discrepancy, which is the topic of our study (under-representation of DNA from dying cells in plasma).

We neglected mitochondrial DNA, as it is not measured in methylation cell-of-origin analysis. Similarly to the argument above, if some of the total DNA measured in plasma is in fact, mitochondrial, this would mean that genomic cfDNA concentration is actually lower than the estimates, meaning that an even smaller fraction of DNA from dying cells is measured in plasma.

**Recommendations For The Authors**

**Reviewer #1 (Recommendations For The Authors):**
I think readers would appreciate the authors commenting or addressing the following points, in addition to addressing the concerns I raised about the methods section in the public review:What variables and considerations did the authors omit in this study?1. Cell-free DNA is found in virtually every biofluid.Thus, the fact that cell-free DNA is not present in the plasma does not mean it cannot be detected elsewhere. This also implies that phagocytosis may not be the only factor related to cfDNA not being present in the blood. One example (of many, many others) is neutrophil-derived cell-free DNA, which is present in the urine.

Indeed, dying cells and their DNA can be consumed locally, released into the blood, or shed outside the body. The latter is a function of tissue topology. For example, intestinal epithelial cell turnover releases material to the lumen of the gut (i.e., stool); kidney and bladder cell turnover releases material to urine; and lung epithelium releases material to the air spaces. In these cases, the absence of cfDNA in plasma is expected. However, in cases where tissue topology dictates release to blood, low representation in cfDNA indicates local consumption or a related mechanism. In Figure 1 of the manuscript, we distinguish between tissues according to their topology, labeling organs that shed material to the outside denoted by open circles.

Neutrophil-derived DNA in urine likely represents a local process in the kidney (neutrophils that penetrate the epithelium and fall into the urine). Neutrophils that die elsewhere in the body must release cfDNA to the blood before it can reach the urine. Hence, quantifying plasma cfDNA is a legitimate approach for assessing the relationship between cell death and cfDNA. The revised text clarifies this point. We made revisions to the initial paragraph in the results section and a paragraph within the discussion to provide clarity on this topic:

“Based on atlases of human cell type-specific methylation signatures, Moss et al. and Loyfer et al. analyzed the main cell types contributing to plasma cfDNA. They found the primary sources of plasma cfDNA to be blood cells: granulocytes, megakaryocytes, macrophages, and/or monocytes (the signature could not differentiate between the last two), lymphocytes, and erythrocyte progenitors. Other cells that had detectable contributions are endothelial cells and hepatocytes. Qualitatively, these cells represent most of the leading cell types in cellular turnover, as shown in Sender & Milo 2021 (Sender and Milo 2021). Epithelial cells of the gastrointestinal tract, lung, kidney, bladder, and skin are other cell types that significantly contribute to cellular turnover. Dying cells in these tissues are shed into the gut lumen, the air spaces, the urine, or out of the skin (note that while DNA from gut, lung, and kidney epithelial cells can be found in stool, bronchoalveolar lavage, and urine, the fate of DNA from skin cells is not known). This arrangement may explain why DNA from these cell types is not represented in plasma cfDNA in healthy conditions. Therefore, it appears that cells with high cfDNA plasma levels are those with relatively high turnover that are not being shed out of the body.”

“A comparison between the different types of cells shows a trend in which less DNA flux from cells with higher turnover gets to the bloodstream. In particular, a tiny fraction (1 in 3x104) of DNA from erythroid progenitors arrives at the plasma, indicating an extreme efficiency of the DNA recovery mechanism. Erythroid progenitors are arranged in erythroblastic islands. Up to a few tens of erythroid progenitors surround a single macrophage that collects the nuclei extruded during the erythrocyte maturation process (pyrenocytes) (Chasis and Mohandas 2008). The amount of DNA discarded through the maturation of over 200 billion erythrocytes per day (Sender and Milo 2021) exceeds all other sources of homeostatic discarded DNA. Our findings indicate that the organization of dedicated erythroblastic islands functions highly efficiently regarding DNA utilization. Neutrophils are another high-turnover cell type with a low level of cfDNA. When contemplating the process of NETosis (Vorobjeva and Chernyak 2020), the existence of cfDNA originating from live neutrophils would potentially diminish the concentration of cfDNA released by dying neutrophils, thereby amplifying the observed ratio for this particular cell type. The overall trend of higher turnover resulting in a lower cfDNA to DNA flux ratio may indicate similar design principles, in which the utilization of DNA is better in tissues with higher turnover. However, our analysis is limited to only several cell types (due to cfDNA test and deconvolution sensitivities), and extrapolation to cells with lower cell turnover is problematic.”

1. Effect of biofluid storage.Cell-free DNA continues to degrade after it is extracted via blood draw. This is not expected to change tissue of origin predictions (although that remains to be shown in the literature), but definitely affects extraction yield. This is not accounted for (or even discussed) in the manuscript. It would be important to understand how this was done for the data presented here.

The paper integrates data from multiple recent studies that adhered to state-of-the-art procedures requiring rapid processing of blood samples. In fact, earlier studies that were not careful to isolate plasma quickly typically reported very high concentrations due to the lysis of leukocytes and artifactual release of genomic DNA. Rapid plasma isolation and DNA extraction typically yield 5ng/ml in healthy donors, as stated in the paper (last paragraph of Results).

1. Batch effectsBatch effects are not discussed here and can affect cfDNA yields.

Our analysis relies on data reported by multiple studies from different groups, which independently results in similar key findings (total concentration of cfDNA and the relative contribution of different tissues). Thus, batch effects are unlikely to affect the calculations markedly.

1. Cell-free DNA extraction kitsDifferent kits and methods extract cell-free DNA at different quantities. Importantly, much research has been done recently that most kits are not sensitive for ultrashort cell-free DNA (of lengths ~50bp). This may represent most of the DNA present in plasma. This raises an important question: are the yields that are being used in Moss et al (where I presume the total concentration is taken from) accurate? Is there more cell-free DNA that was missed? While the importance of this ultrashort cfDNA has yet to be shown, it is in the blood. Thus, the authors' model may underestimate ratios by not accounting for this. This is mentioned in the discussion, but it is not evident why it was not added into the model.

The Qiagen cfDNA extraction kit can detect 50bp fragments. As shown in the specification sheets of the kit(available here), urine DNA contains abundant DNA fragments that peak at 50bp. In contrast, plasma cfDNA does not contain such fragments at appreciable concentrations. This suggests that small fragments, 50-150bp long, are not a major component of cfDNA, and thus, our measurements of the total concentration of cfDNA are not dramatically underestimated.

The convention regarding the size distribution of cfDNA fragments is based on extensive evidence using multiple approaches. For example, a study that profiled the DNA released by multiple cell lines in vitro (Aucamp et al. 2017) used another kit for DNA isolation – the NucleoSpin Gel and PCR Clean-up kit (Macherey-Nagel, Düren, Germany). This kit does extract fragments that are 50bp long (nucleospin-gel-and-pcr-clean-up-mini). Indeed, the DNA released from cultured cells did contain a peak at 50bp, but it was minor compared with the nucleosome-size peak.

More recently, several studies did suggest the presence of ultra-short cfDNA fragments, 50 bp long on average, and concluded that such fragments might be present at a molar concentration that is comparable to that of nucleosome-protected DNA (for example, (Hisano et al. 2021)).

Thus, our model estimates can be off by up to 2-fold (that is, actual cfDNA concentration measured in most studies overlooks the small fragments and thus underestimates the actual concentration of cfDNA by 2-fold). This is incorporated into the revised manuscript.

We note that we cannot exclude the presence of abundant ultra-short DNA fragments (e.g., 10bp long). However, such fragments are not measurable in cfDNA analysis. Thus, we can refine our conclusion and state that only a small fraction of DNA of dying cells appears as measured cfDNA.We included a section in the methods detailing the integration of a potential factor for the short fragments and revised the discussion:

“The overall plasma cfDNA concentration was multiplied by a factor of 1.5 to accommodate for the presence of small fragments of approximately 50 base pairs of cfDNA in the plasma. These fragments are suggested to contribute comparable molar concentrations (Hisano, Ito, and Miura 2021). Despite having approximately one-third of the mass, it is reasonable to presume that these fragments represent a similar number of genomes. This assumption is based on the idea that their source is a broken nucleosome unit, and the fragments represent the portion that was not degraded. Given the restricted data and its interpretation, we consider factors spanning the range of 1 (negligible effect) and 2 (doubling of the amount). The chosen factor, 1.5, is selected as the midpoint within this range of uncertainty.”

“In this study, we report a surprising, dramatic discrepancy between the measured levels of cfDNA in the plasma and the potential DNA flux from dying cells. One hypothetical explanation for that discrepancy is the limited sensitivity of typical cfDNA assays to short DNA fragments, which may contribute a significant fraction of the overall cfDNA mass. Regular cfDNA analysis shows a size distribution concentrated around a length of 165 base pairs (bp). The sizes in ctDNA vary more, but most are longer than 100 bp (Alcaide et al. 2020; Udomruk et al. 2021). Recent studies suggested a significant fraction of single-strand ultrashort fragments (length of 25-60 bp) (Cheng et al. 2022; Hisano, Ito, and Miura 2021). However, the total amount of DNA contained in these fragments is less than or comparable to that of the longer “regular” nucleosome-protected cfDNA fragments (Cheng et al. 2022; Hisano, Ito, and Miura 2021), arguing against ultrashort fragments as a dominant explanation for the “missing” cfDNA material. We integrated the estimate provided by Hisano et al. into our analysis as a modifying factor for both the total concentration and uncertainty of plasma cfDNA. Importantly, this incorporation did not alter the overall conclusions, as the discrepancy between the cfDNA plasma concentration and potential DNA flux remains on the same order of magnitude. We note that we cannot exclude the presence of abundant DNA fragments that are even shorter (e.g., 10bp long) and are not measurable in cfDNA analysis. Thus, our formal conclusion is that only a small fraction of the DNA of dying cells appears as measurable cfDNA.”

1. Health status of samples analyzed.Health, sex and physical activity affects cfDNA yields. This is not accounted for or discussed in the manuscript.

We incorporated several enhancements to improve our analysis in response to the provided feedback. In our revised examination, we drew upon the total plasma concentration of cfDNA, as documented in a study conducted by (Meddeb et al. 2019), while considering the influence of age and sex on these concentrations. To ensure the cohort's alignment, we focus on relatively young and healthy individuals, specifically those below the age of 47. This approach allowed for a more meaningful comparison with the estimated DNA flux from a reference male human aged between 20 and 30 years.

Furthermore, we factored in the expected variations stemming from sex, age, and other relevant factors, as elucidated in the works of (Meddeb et al. 2019; Madsen et al. 2019). Our intent in doing so was to demonstrate that these factors are unlikely to alter our conclusions substantially when considering a healthy population. We summarize the changes in the first paragraph, replacing the “Tissue-specific cfDNA concentration” subsection of the method, and the fourth paragraph added to the discussion:

“Our estimates for total plasma cfDNA concentration were derived from the median concentration observed in individuals below 47 years of age (n=52), as reported by (Meddeb et al. 2019). To complement this, we integrated our total concentration estimates with data on the proportion of cfDNA originating from specific cell types, leveraging a plasma methylome deconvolution method described by (Loyfer et al. 2023), which did not provide absolute quantities of cfDNA.To quantify the uncertainty associated with our cfDNA concentration estimates, we employed a methodology that considered several sources of variation. First, we incorporated the confidence interval of the median concentration reported by Meddeb et al. as a measure of uncertainty. Additionally, we accounted for individual-specific and analytic variations based on the study by (Madsen et al. 2019), encompassing factors such as the precise timing of measurements and assay precision. These sources of uncertainty were combined using the approach outlined below.”

“Our current analysis focused on estimating plasma cfDNA concentration and cellular turnover in a cohort of healthy, relatively young individuals. The total plasma cfDNA concentrations were sourced from healthy individuals below 47 years, as reported by (Meddeb et al. 2019). We use data analyzed based on plasma samples from healthy individuals to estimate the proportion of cfDNA originating from specific cell types (Loyfer et al. 2023). These values were then compared to the potential DNA flux resulting from homeostatic cellular turnover, estimated for reference healthy males aged between 20 and 30 (Sender and Milo 2021). In our analysis, we considered various sources of uncertainty, including inter-individual variation, variability in the timing of sample collection, and analytical precision (Madsen et al. 2019; Meddeb et al. 2019). These factors collectively contributed to an uncertainty factor of less than 3. Importantly, this level of uncertainty does not alter our conclusion regarding the relatively small fraction of DNA present in plasma as cfDNA. Furthermore, we acknowledge that age and sex can impact total cfDNA concentration, as demonstrated by (Meddeb et al. 2019), with potential variations of up to 30%. However, as the results of our analysis present a much larger difference, these effects do not change the conclusions drawn from our analysis. Nevertheless, age and health status may influence the proportion of cfDNA originating from specific cell types and their corresponding cellular turnover rates. Consequently, the ratios themselves may vary in the elderly population or individuals with underlying health conditions.”

**Reviewer #2 (Recommendations For The Authors):**
1. Align the cohorts to estimate DNA production and plasma cfDNA levels. Cellular turnover rate and plasma cfDNA levels vary with age, sex, circadian clock, and other factors (Madsen AT et al, EBioMedicine, 2019). This study estimated DNA production using data abstracted from a homogenous group of healthy control males (Sender & Milo, Nat Med 2021). On the other hand, plasma cfDNA levels were obtained from datasets of more diverse cohort of healthy males and females with a wide range of ages (Loyfer et al. Nature, 2023 and Moss et al., Nat Commun, 2018).

We have incorporated several enhancements to improve the coherence of our analysis. In our revised examination, we drew upon the total plasma concentration of cfDNA, as documented in a study conducted by (Meddeb et al. 2019), while considering the influence of age and sex on these concentrations. To ensure the cohort's alignment, we focus on relatively young and healthy individuals, specifically those below the age of 47. This approach allowed for a more meaningful comparison with the estimated DNA flux from a reference male human aged between 20 and 30 years.

There was no specific estimate for a cohort of young males in both Meddeb et al. and Loyfer et al.; however, we factored in the expected variations stemming from sex, age, and other relevant factors, as elucidated in literature (Meddeb et al. 2019; Madsen et al. 2019). Thus, we demonstrate that sex and age have a small effect on the cfDNA concentrations and thus are unlikely to alter our conclusions substantially when considering a healthy population.

We summarize the changes in the first paragraph, replacing the “Tissue-specific cfDNA concentration” subsection of the method, and the fourth paragraph added to the discussion.

“Our estimates for total plasma cfDNA concentration were derived from the median concentration observed in individuals below 47 years of age (n=52), as reported by (Meddeb et al. 2019). To complement this, we integrated our total concentration estimates with data on the proportion of cfDNA originating from specific cell types, leveraging a plasma methylome deconvolution method described by (Loyfer et al. 2023), which did not provide absolute quantities of cfDNA.To quantify the uncertainty associated with our cfDNA concentration estimates, we employed a methodology that considered several sources of variation. First, we incorporated the confidence interval of the median concentration reported by Meddeb et al. as a measure of uncertainty. Additionally, we accounted for individual-specific and analytic variations based on the study by (Madsen et al. 2019), encompassing factors such as the precise timing of measurements and assay precision. These sources of uncertainty were combined using the approach outlined below.”

“Our current analysis focused on estimating plasma cfDNA concentration and cellular turnover in a cohort of healthy, relatively young individuals. The total plasma cfDNA concentrations were sourced from healthy individuals below 47 years, as reported by (Meddeb et al. 2019). We use data analyzed based on plasma samples from healthy individuals to estimate the proportion of cfDNA originating from specific cell types (Loyfer et al. 2023). These values were then compared to the potential DNA flux resulting from homeostatic cellular turnover, estimated for reference healthy males aged between 20 and 30 (Sender and Milo 2021). In our analysis, we considered various sources of uncertainty, including inter-individual variation, variability in the timing of sample collection, and analytical precision (Madsen et al. 2019; Meddeb et al. 2019). These factors collectively contributed to an uncertainty factor of less than 3. Importantly, this level of uncertainty does not alter our conclusion regarding the relatively small fraction of DNA present in plasma as cfDNA. Furthermore, we acknowledge that age and sex can impact total cfDNA concentration, as demonstrated by (Meddeb et al. 2019), with potential variations of up to 30%. However, as the results of our analysis present a much larger difference, these effects do not change the conclusions drawn from our analysis. Nevertheless, age and health status may influence the proportion of cfDNA originating from specific cell types and their corresponding cellular turnover rates. Consequently, the ratios themselves may vary in the elderly population or individuals with underlying health conditions.”

1. "cfDNA fragments are not created equal". Recent studies demonstrate that cfDNA composition vary with disease state. For example, cfDNA GC content, fraction of short fragments, and composition of some genomic elements increase in heart transplant rejection compared to no-rejection state (Agbor-Enoh, Circulation, 2021). The genomic location and disease state may therefore be important factors to consider in these analyses.

In this study, we addressed the total amount of cfDNA in healthy individuals without regard to GC content, representation of different genomic regions, or fragment length, as the goal was to understand if cell death rates are fully accounted for by cfDNA concentration. We agree that it will be interesting to study the relative representation of the genome in cfDNA and the processes that determine cfDNA concentration in pathologies beyond the rate of cell death. These topics for future research fall beyond this study's scope.

1. Alternative sources of DNA production should be considered. Aside from cell death, DNA can be released from cells via active secretion. This and other additional sources of DNA should be considered in future studies. The distinct characteristics of mitochondrial DNA to genomic DNA should also be considered.

We know only a few specific cases whereby DNA is released from cells that are not dying. These include the release of DNA from erythroblasts and megakaryocytes to generate anucleated erythrocytes and platelets (Moss et al. 2022, cited in our paper) and the release of NETs from neutrophils.

The presence of cfDNA fragments originating from megakaryocytes and erythroblasts indicates the elimination of megakaryocytes and erythroblasts and the birth of erythrocytes and platelets. However, the considerations in the rest of the paper still apply: the concentration of cfDNA from these sources is far lower than expected from the cell turnover rate.

Concerning NETosis: the presence of cfDNA originating in neutrophils that have not died would reduce the concentration of cfDNA from dying neutrophils and thus further increase the discrepancy, which is the topic of our study (under-representation of DNA from dying cells in plasma).

We updated a paragraph in the discussion regarding this issue:

“A comparison between the different types of cells shows a trend in which less DNA flux from cells with higher turnover gets to the bloodstream. In particular, a tiny fraction (1 in 3x104) of DNA from erythroid progenitors arrives at the plasma, indicating an extreme efficiency of the DNA recovery mechanism. Erythroid progenitors are arranged in erythroblastic islands. Up to a few tens of erythroid progenitors surround a single macrophage that collects the nuclei extruded during the erythrocyte maturation process (pyrenocytes) (Chasis and Mohandas 2008). The amount of DNA discarded through the maturation of over 200 billion erythrocytes per day (Sender and Milo 2021) exceeds all other sources of homeostatic discarded DNA. Our findings indicate that the organization of dedicated erythroblastic islands functions highly efficiently regarding DNA utilization. Neutrophils are another high-turnover cell type with a low level of cfDNA. When contemplating the process of NETosis (Vorobjeva and Chernyak 2020), the existence of cfDNA originating from live neutrophils would potentially diminish the concentration of cfDNA released by dying neutrophils, thereby amplifying the observed ratio for this particular cell type. The overall trend of higher turnover resulting in a lower cfDNA to DNA flux ratio may indicate similar design principles, in which the utilization of DNA is better in tissues with higher turnover. However, our analysis is limited to only several cell types (due to cfDNA test and deconvolution sensitivities), and extrapolation to cells with lower cell turnover is problematic.”

We neglected mitochondrial DNA, as it is not measured in methylation cell-of-origin analysis. Similarly to the argument above, if some of the total DNA measured in plasma is in fact mitochondrial, this would mean that genomic cfDNA concentration is actually lower than the estimates, meaning that an even smaller fraction of DNA from dying cells is measured in plasma.